

# Exploring influences on radiation protection compliance: a directed acyclic graph-based cross-sectional study in a non-teaching hospital in western China

Lulu Cao[1,*], Jinwen Wu[1,*], Li Jiang[1], Yi Tao[1], Huaping Huang[2] and XiaoJing Xue[3]

[1] Department of Radiology, Mianyang Central Hospital, School of Medicine, University of Electronic Science and Technology of China, Mianyang, Sichuan, China
[2] Department of Nursing, Mianyang Central Hospital, School of Medicine, University of Electronic Science and Technology of China, Mianyang, Sichuan, China
[3] Department of Radiation Oncology Center, Mianyang Central Hospital, School of Medicine, University of Electronic Science and Technology of China, Mianyang, Sichuan, China
[*] These authors contributed equally to this work.

## ABSTRACT

**Background**. Occupational radiation exposure poses significant health risks to medical personnel. However, the causal pathways linking protective knowledge, attitudes, and behavior (KAB) remain underexplored. Therefore, this study aimed to apply directed acyclic graphs (DAGs) to clarify the mechanistic relationships among these factors.

**Methods**. A cross-sectional survey of 335 radiation workers from a non-teaching level III general hospital in western China used validated scales to measure KAB. DAGs were constructed based on theoretical frameworks and previous evidence, complemented by correlation analyses, multivariate linear regression, and structural equation modeling.

**Results**. Radiation protection knowledge exerted the strongest direct effect on protective behavior ($\beta = 0.555$, $p < 0.001$). Attitude mediated 18.3% of the total effect ($\beta = 0.024$, 95% confidence interval [0.003–0.045]). Age was a significant negative predictor of compliance ($\beta = -0.390$, $p < 0.001$), while training improved both knowledge ($\beta = 0.394$, $p < 0.001$) and behavior ($\beta = 0.147$, $p < 0.001$). Educational level was significantly correlated with knowledge acquisition ($\beta = 0.101$, $p = 0.029$) but did not directly influence behavior. Participants demonstrated positive attitudes (mean = 21.35/25) and behaviors (mean = 28.99/35). However, critical knowledge gaps persisted in radiation culture (28% correct) and emergency protocols (25% correct).

**Conclusions**. This study applied DAGs to clarify causal mechanisms in radiation protection, highlighting knowledge acquisition as a key driver of safe practices. Age-specific interventions and standardized training programs are recommended to address knowledge deficits and mitigate age-related behavioral decline. These findings provide a methodological foundation for optimizing occupational health strategies, with implications for policy design and future longitudinal validation.

Corresponding authors
Huaping Huang,
Jrzhou26@aliyun.com
XiaoJing Xue, 1250312090@qq.com

## INTRODUCTION

Radiological examinations have become an important tool in clinical diagnosis due to advancements in medical technology and social progress. While these examinations provide patients with valuable diagnostic information, they concurrently increase their exposure to radiation (*Yashima & Chida, 2022*). According to the World Health Organization, more than 4.2 billion diagnostic radiological examinations, 40 million nuclear medicine imaging procedures, and 8.5 million radiation treatments are conducted globally each year (*World Health Organization, 2016*). Consequently, the application of ionizing radiation inspection should be optimized while strengthening radiation protection measures to minimize related health hazards. Given the improvements in inspection technology and protective measures, the radiation dose received during radiological examinations is generally within the safe range (*Giammarile et al., 2024*).

Despite existing protective measures, prolonged radiation exposure among medical personnel and individual sensitivity can still result in tissue and organ damage, as well as physiological dysfunction (*Baudin et al., 2021*). Therefore, clinicians and radiologists should utilize personal protective measures during radiological diagnosis and treatment to minimize radiation hazards (*Awadghanem et al., 2020*).

The detrimental effects of ionizing radiation on the human body are generally categorized into deterministic and stochastic effects. Deterministic effects, such as acute radiation sickness and radiation-induced skin injury, occur when exposure to ionizing radiation exceeds a certain threshold, with the severity of damage increasing at higher doses. In contrast, stochastic effects, such as carcinogenic and genetic outcomes, do not have a threshold dose; instead, their probability of occurrence is related to the dose size (*Shafiq & Mehmood, 2024*; *Talapko et al., 2024*). While eliminating stochastic effects is impossible, they can be reduced by implementing appropriate protective measures.

Both acute and chronic exposure to ionizing radiation have been shown to adversely affect human health, with considerable variation in the radiation sensitivity of different human tissues (*Britel, Bourguignon & Foray, 2018*). Healthcare personnel exposed to ionizing radiation have an increased incidence of developing various tumors and leukemia. Low radiation doses have been linked to an increased risk of cancer. Its prolonged exposure has also been found to affect the motility, normal morphology, and DNA fragmentation index of sperm in Chinese men (*Kumar et al., 2014*; *Zhou et al., 2016*).

Considering these findings, the International Commission on Radiological Protection (ICRP) reduced the occupational radiation limit for radiological workers from 50 mSv/year to an average of 20 mSv/year over five consecutive years, with a maximum of 50 mSv allowed in any single year (*Frane & Bitterman, 2022*). Therefore, current protection levels are based on the principle that, while radiation risk cannot be eliminated, it can be kept at a tolerable level. The ICRP proposes three fundamental principles of radiation protection as follows: "justification", "optimization", and "dose limitation" (*Yeung, 2019*).

Nevertheless, the level of attention paid to radiation protection among radiation workers with varying levels of knowledge differs considerably. Some practice excessive protection—usually driven by an overestimation of radiation risks—while others insufficiently use

protective equipment, leading to excessive exposure and potential health risks (*Alyousef et al., 2023*). Studies consistently show that individuals with greater knowledge of radiation protection are more likely to implement effective protective measures. This underscores the critical role of radiation protection education (*Rahimi et al., 2021*; *Jeyasugiththan et al., 2023*; *Salih et al., 2023*).

An individual's knowledge of radiation protection significantly influences their attitudes and behaviors, which consequently impact their health outcomes (*Partap et al., 2019*).

In China, where the number of radiological workers in medical institutions exceeds 450,000 and continues to increase annually (*Deng & Sun, 2021*), understanding the determinants of radiation protection behavior is essential. However, research in this area—particularly regarding the causal mechanisms underlying these behaviors—remains limited.

While cross-sectional studies have effectively identified associations between knowledge, attitudes, and behaviors (KAB) in radiation protection, they are inherently limited in establishing causal relationships. To address this gap, this study introduces a directed acyclic graph (DAG) framework, which is a robust tool for formalizing causal assumptions based on theoretical models and prior evidence (*Akinkugbe et al., 2016*; *Textor et al., 2016*). By constructing a DAG, we aimed to clarify whether radiation protection knowledge directly influences protective behaviors or operates indirectly through attitudinal mediation, while also accounting for potential confounders, such as age, educational level, and training exposure. This approach strengthens the rigor of causal inference and aligns with recent recommendations for integrating advanced causal methods into observational studies in occupational health (*Tennant et al., 2021*; *Zheng et al., 2023*). Through this innovative framework, we seek to generate actionable insights to inform and improve radiation protection practices and policies among Chinese healthcare workers.

## MATERIALS & METHODS

### Participants and procedure

This cross-sectional survey was conducted at a non-teaching Level III general hospital (2,200 beds) in western China, which serves as a regional referral center for a population exceeding 5 million. Convenience sampling—a non-probability sampling method—was adopted. A questionnaire, administered in Chinese, was distributed to participants who met the following inclusion criteria: age >18 years and active employment status as radiation workers in China. Participants included multi-disciplinary clinical staff routinely exposed to ionizing radiation, encompassing physicians, nurses, radiological technicians, and medical physicists. Of the 350 questionnaires distributed, 335 were returned, yielding an effective recovery rate of 95.7%.

### Research instruments

The current questionnaire was designed based on existing research and previously developed questionnaires, including the Training Requirements for Radiation Protection for Medical Radiation Workers, Basic Standards for Ionizing Radiation Protection and Radiation Source Safety, and Requirements for Radiation Protection in Diagnostic
Radiology (*Han et al., 2012*; *Partap et al., 2019*; *Schroderus-Salo et al., 2019*; *Chatzis et al., 2021*). After the members of the research group discussed and modified the questionnaire items, six experts (including two senior radiation safety officers, two radiologists with >20 years of clinical experience, one occupational health epidemiologist, and one nursing administrator) were invited to review the prepared items individually. A formal questionnaire was finalized after careful revision based on their feedback.

The questionnaire consisted of four sections. Specifically, the first section collected general information, including the participant's gender, age, educational level, profession, work unit, and daily exposure time. The second section assessed radiation protection knowledge through 14 questions, including the difference between ionizing and non-ionizing radiation, three principles of radiation protection, the correct use of radiation protection equipment, radiation protection culture, health examinations of radiation workers, emergency management, and radiation dose limits. Responses indicating "yes" were assigned 1, while those indicating "no" or "do not know" were scored as 0. Furthermore, the total scores ranged from 0 to 14 points, with higher scores indicating greater knowledge.

The third section inquired about attitudes toward five items as follows: the accuracy and reliability of personal dosimeters, danger of radiation to health, need to standardize protective equipment use, regular provision of radiation protection knowledge and training, and necessity of radiation protection measures in daily diagnosis and treatment. Responses were rated on a 5-point Likert scale (1 = strongly disagree; 5 = strongly agree), with total scores ranging from 5 to 25 (*Jebb, Ng & Tay, 2021*). Higher scores indicated a more positive attitude toward radiation protection. The fourth section assessed behavior and included seven items related to the implementation of protective measures by radiation workers. Responses were rated on a 5-point Likert scale (1 = never; 5 = always). Scores ranged from 7 to 35, with higher scores indicating better radiation protection behavior. The abovementioned six experts established content validity using a 4-point relevance scale (1 = irrelevant; 4 = essential). Items with a Content Validity Index (CVI) <0.78 were discarded. Face validity was established through cognitive interviews with 15 radiation workers (>10 years of clinical experience) from diverse clinical roles, which confirmed the clarity and contextual appropriateness of the items. Construct validity was verified through exploratory factor analysis, showing sampling adequacy (Kaiser–Meyer–Olkin measure = 0.883) and significant sphericity (Bartlett's test $\chi^2$ = 6019.1, $p < 0.001$). Cronbach's alpha coefficients for KAB were 0.921, 0.878, and 0.743, indicating good internal consistency (*Bujang, Omar & Baharum, 2018*).

## Defining covariates for a DAG

A DAG is a visual representation that illustrates causality between variables, comprising nodes (which represent variables) and directed edges connecting nodes (representing causality between variables) (*Tennant et al., 2021*). Our research is based on theoretical frameworks (*e.g.*, the KAB model), empirical evidence, and domain expertise to formalize hypothesized causal relationships among variables. Nodes included demographic factors (such as age, sex, educational level, marital status, and number of children), occupational

factors (including profession, working unit, career duration, daily exposure time, and radiation protection training), health-related factors (such as subjective health status), and core variables (including radiation protection knowledge and attitude). Arrows were oriented according to previous literature: age directly influenced both knowledge (*via* cumulative experience) and behavior (*via* potential burnout); educational level and training were linked to knowledge acquisition and subsequent behavior; and knowledge was modeled to affect behavior both directly and through attitude mediation, as supported by the KAB theory.

Non-modifiable variables (*e.g.*, sex and marital status) were specified as confounders with direct paths to behavior, while subjective health status mediated the impact of occupational exposure on compliance. The DAG adhered to acyclicity constraints, with radiation protection behavior designated as the sink node and non-modifiable factors as source nodes.

## Data analysis

All data were imported into Microsoft Excel 2019 to create a database. Statistical analyses were performed using IBM SPSS Statistics for Windows, version 25.0 (IBM Corp., Armonk, N.Y., USA). Quantitative data were presented as means and standard deviations. An independent sample $t$-test and a one-way analysis of variance were used for inter-group and multi-group comparisons, respectively. Scheffe's test was used for *post-hoc* analysis. Qualitative count data were presented as composition ratios and rates. Pearson's correlation coefficient was used to analyze the relationships between radiation protection KAB. Multivariate linear regression analysis was conducted to examine influencing factors, with a statistical significance threshold set at $p < 0.05$.

A DAG was constructed using the DAGitty package (version 3.0) in R Studio (http://www.dagitty.net) to formalize causal assumptions and address potential confounding. Causal assumptions were formalized using DAGitty (v3.0). The tool's d-separation function verified implied conditional independencies (*e.g.*, Attitude $\perp$ Radiation Training | Knowledge), while its adjustment set generator identified minimally sufficient covariates (*e.g.*, {Age, Education level} for the Knowledge→Behavior path). These steps ensured DAG's consistency prior to structural equation modeling.

## Ethical considerations

The study protocol was approved by the Biomedical Ethics Committee of Mianyang Centre Hospital (approval number: s20240372-01) and registered in the Chinese Clinical Trials Registry (registration number: ChiCTR2400084431) prior to study initiation. Before completing the survey, all participants provided informed consent through an online electronic consent form embedded in the questionnaire platform. Data collection remained strictly confidential and anonymous. In accordance with the ethical standards of the Declaration of Helsinki, participants were informed of their right to withdraw from the study or terminate participation at any phase without any disadvantages. Completion of the questionnaires required approximately 10 min.

## RESULTS

### Participant characteristics

Among the 335 medical workers, 231 (69.0%) were males, 148 (44.2%) were aged 30–40 years, and 210 (62.7%) had obtained a bachelor's degree. A total of 138 (41.2%) and 102 (30.4%) were nurses and doctors, respectively. The largest proportion of medical personnel worked in the radiology department (113 individuals, 33.7%). Additionally, the majority of medical workers (241, 71.9%) were exposed to radiation for <4 h per day. The subjective health status of 184 (54.9%) participants was "good". Overall, 140 (41.8%) participants had no children, while 216 (64.5%) were married. Table 1 presents the participant characteristics.

### Knowledge, attitudes, and behaviors

Radiological workers' knowledge of radiation protection ranged from 0 to 14 points, with an average score of 8.65 ($\pm$4.63). Radiation protection attitudes ranged from 15 to 25 points, with a mean score of 21.35 ($\pm$2.45). Additionally, radiation protection behavior was scored on a scale of 7–35, with a mean score of 28.99 ($\pm$5.85). Table 2 presents the results for radiation protection KAB.

### Correlations between knowledge, attitudes, and behaviors

Radiation protection knowledge was significantly positively correlated with attitude ($r = 0.256$, $p < 0.01$) and behavior ($r = 0.651$, $p < 0.01$). Additionally, radiation protection attitude was significantly positively correlated with behavior ($r = 0.259$, $p < 0.01$; Table 3).

### Factors affecting radiation protection behaviors

A multiple linear regression analysis was conducted, with the radiation protection behavior score as the dependent variable and factors with statistical significance in the univariate analysis as independent variables. The results indicated that radiation protection behaviors were affected by age ($t = -5.085$, $p < 0.001$), educational level ($t = 2.188$, $p = 0.029$), radiation protection training ($t = 3.824$, $p < 0.001$), number of children ($t = 2.156$, $p = 0.032$), knowledge about radiation protection ($t = 10.989$, $p = 0.032$), and attitude toward radiation protection ($t = 2.086$, $p = 0.038$; Table 4). Furthermore, the total explanatory power of these factors for radiation protection behavior was 51.6% ($F = 30.655$, $p < 0.001$).

### DAG analysis

To address potential confounding and formalize causal assumptions, the DAG was constructed using variables identified as statistically significant in both univariate and multivariate analyses, including age, educational level, radiation protection training, number of children, knowledge about radiation protection, and attitude toward radiation protection. The DAG framework hypothesized causal pathways grounded in the KAB theory and supported by empirical evidence from occupational health studies.

Key pathways demonstrated that higher radiation protection knowledge directly predicted better compliance behaviors ($\beta = 0.551$, $p < 0.001$), meaning each standard deviation increase in knowledge corresponded to a 55% improvement in protective

**Table 1  Participant characteristics ($n = 335$).**

| Characteristics | Categories | n (%) | M ± SD | Statistic (Test) | P-value[†] |
|---|---|---|---|---|---|
| Gender | Female | 104 (31.0) | 30.45 ± 5.12 | 3.008 (t) | 0.003 |
| | Male | 231 (69.0) | 28.42 ± 6.13 | | |
| Age (years) | <30 | 109 (32.5) | 29.91 ± 5.02 | 3.625 (F) | 0.016 |
| | 30–40 | 148 (44.2) | 29.62 ± 4.93 | | |
| | 41–50 | 54 (16.1) | 27.26 ± 7.12 | | |
| | >50 | 24 (7.2) | 25.64 ± 8.93 | | |
| Educational level | College and below | 16 (4.8) | 21.25 ± 8.35 | 8.560 (F) | 0.001 |
| | Bachelor | 210 (62.7) | 29.12 ± 5.57 | | |
| | ≥Master | 109 (32.5) | 30.02 ± 5.32 | | |
| Marital status | Unmarried | 115 (34.2) | 30.49 ± 4.23 | 7.118 (F) | 0.016 |
| | Married | 216 (64.5) | 28.18 ± 6.45 | | |
| | Divorced | 4 (1.2) | 30.00 ± 3.46 | | |
| Number of children | None | 140 (41.8) | 30.29 ± 4.36 | 6.899 (F) | 0.002 |
| | 1 | 155 (46.3) | 28.13 ± 6.45 | | |
| | ≥2 | 40 (11.9) | 27.80 ± 7.12 | | |
| Profession | Doctor | 102 (30.4) | 30.4 ± 4.23 | 31.679 (F) | <0.001 |
| | Technicians | 64 (19.1) | 32.66 ± 2.93 | | |
| | Nurse | 138 (41.2) | 26.7 ± 6.1 | | |
| | The other health personnel[*] | 31 (9.3) | 27.1 ± 8.5 | | |
| Working unit | Radiology department | 113 (33.7) | 30.02 ± 5.74 | 9.760 (F) | <0.001 |
| | Department of interventional surgery | 36 (10.7) | 29.72 ± 3.84 | | |
| | Nuclear medicine department | 30 (9.0) | 32.32 ± 3.44 | | |
| | Endoscopic center | 28 (8.4) | 28.85 ± 4.95 | | |
| | Surgery department | 76 (22.7) | 25.73 ± 6.36 | | |
| | Oncology department | 52 (15.5) | 29.44 ± 6.55 | | |
| Career in present unit (years) | <5 | 116 (34.6) | 29.92 ± 4.52 | 6.439 (F) | <0.001 |
| | 5–10 | 108 (32.2) | 30.28 ± 4.53 | | |
| | 11–20 | 74 (22.1) | 25.92 ± 7.88 | | |
| | >20 | 37 (11.0) | 28.82 ± 6.67 | | |
| Daily exposure time (hours) | <4 | 241 (71.9) | 28.57 ± 6.12 | 4.733 (F) | 0.014 |
| | 4–8 | 77 (23.0) | 30.64 ± 4.62 | | |
| | >8 | 17 (5.1) | 28.61 ± 6.75 | | |
| Radiation protection training | Yes | 173 (51.6) | 30.94 ± 4.89 | −6.625 (t) | <0.001 |
| | No | 162 (48.4) | 26.91 ± 6.41 | | |
| Subjective health status | Bad | 8 (2.4) | 23.00 ± 9.56 | 2.812 (F) | 0.086 |
| | Average | 143 (42.7) | 28.57 ± 5.58 | | |
| | Good | 184 (54.9) | 29.58 ± 5.72 | | |

**Notes.**

[†] P-values reflect differences in mean radiation protection behavior scores (M ± SD) across subgroups. Independent samples t-test was used for variables with two categories (e.g., Gender); one-way ANOVA for variables with ≥3 categories (e.g., age, education Level).

[*] The other health personnel: Medical Physicist, Industrial pharmacist.

**Table 2  Degrees of knowledge about radiation protection, attitude towards radiation protection and radiation protection behaviors ($n = 335$).**

| Variable | Mean (SD) | Min~Max | Range |
|---|---|---|---|
| Knowledge about radiation protection | 8.65 (4.63) | 0–14 | 0–14 |
| Attitude towards radiation protection | 21.35 (2.45) | 14–25 | 5–25 |
| Radiation protection behaviors | 28.99 (5.85) | 7–35 | 7–35 |

**Table 3  Correlations of variables ($n = 335$).**

| Variable | Knowledge about radiation protection | Attitude towards radiation protection | Radiation protection behaviors |
|---|---|---|---|
| Knowledge about radiation protection | 1 | | |
| Attitude towards radiation protection | 0.256[**] | 1 | |
| Radiation protection behaviors | 0.651[**] | 0.259[**] | 1 |

Notes.
[**]$P < 0.01$.

**Table 4  Factors affecting radiation protection behaviors ($n = 335$).**

| Variable | B | SE | Beta | t | P |
|---|---|---|---|---|---|
| Constant | 20.289 | 2.756 | / | 7.362 | <0.001 |
| Gender | 0.398 | 0.532 | 0.032 | 0.748 | 0.455 |
| Age (years) | −0.246 | 0.048 | −0.390 | −5.085 | <0.001 |
| Educational level | 1.079 | 0.493 | 0.101 | 2.188 | 0.029 |
| Profession | 0.055 | 0.281 | 0.009 | 0.197 | 0.844 |
| Working unit | 0.002 | 0.133 | 0.001 | 0.011 | 0.991 |
| Marital status | −0.890 | 0.811 | −0.073 | −1.098 | 0.273 |
| Career in present unit (years) | 0.078 | 0.050 | 0.114 | 1.554 | 0.121 |
| Daily exposure time (hours) | −0.598 | 0.428 | −0.058 | −1.398 | 0.163 |
| Radiation protection training | 1.953 | 0.511 | 0.167 | 3.824 | <0.001 |
| Number of children | 1.184 | 0.549 | 0.136 | 2.156 | 0.032 |
| Knowledge about radiation protection | 0.700 | 0.064 | 0.555 | 10.989 | <0.001 |
| Attitude towards radiation protection | 0.204 | 0.098 | 0.086 | 2.086 | 0.038 |

Notes.
$R^2 = 0.533$, Adjusted $R^2 = 0.516$, $F = 30.655$, $p < 0.001$.

practices. Conversely, older age directly reduced compliance ($\beta = -0.260$, $p < 0.001$), indicating 26% lower adherence independent of knowledge/attitude. Radiation protection training simultaneously enhanced both knowledge ($\beta = 0.394$, $p < 0.001$) and compliance ($\beta = 0.147$, $p < 0.001$). Knowledge indirectly influenced behavior through attitude mediation, with an indirect effect accounting for 18.3% of the total effect ($\beta = 0.024$, 95% confidence interval (CI) [0.003–0.045]).

Variables such as educational level ($\beta = 0.101$, $p = 0.029$) and number of children ($\beta = 0.136$, $p = 0.032$) were identified as confounders, requiring adjustment in the multivariate models. Comparison with regression analysis confirmed alignment. Age and radiation protection training remained significant predictors ($p < 0.05$), while the effect

**Table 5 DAG-guided direct and indirect effects on radiation protection behavior.**

| Path | Standardized β† | 95% CI | *p*-value |
|---|---|---|---|
| Direct effects | | | |
| Knowledge → Behavior | 0.551 | [0.447, 0.655] | <0.001 |
| Age → Behavior | −0.260 | [−0.316, −0.204] | <0.001 |
| Training → Knowledge | 0.394 | [0.302, 0.486] | <0.001 |
| Training → Behavior | 0.147 | [0.052, 0.242] | <0.001 |
| Indirect effect | | | |
| Knowledge → Attitude → Behavior | 0.024 | [0.003, 0.045] | 0.026 |

**Notes.**
†Standardized β for the indirect effect is the product of the path coefficients (β Knowledge→Attitude × β Attitude→Behavior) and represents the mediated effect size.

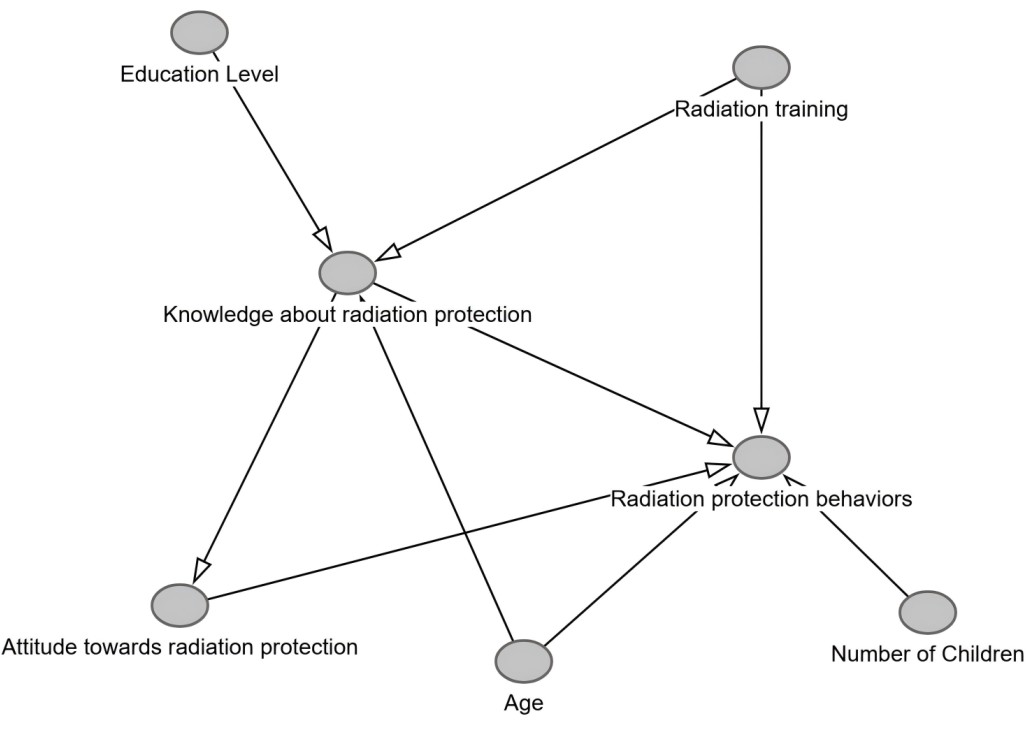

**Figure 1 DAG of radiation protection pathways.**

of educational level was attenuated post-adjustment, indicating confounding. The DAG's logical consistency was verified using the DAGitty package, an open-source causal graph modeling tool for constructing, analyzing and validating DAGs, whose functions include identifying confounding variables, generating adjustment sets, and testing conditional independence hypotheses. However, unmeasured confounders (*e.g.*, workplace culture) and the cross-sectional design limited causal certainty (Table 5, Fig. 1).

## DISCUSSION

The occupational health risks associated with prolonged exposure to low-dose ionizing radiation among medical personnel necessitate rigorous scientific inquiry and policy action. While China has established regulatory frameworks to address radiation-related occupational diseases (*Jianbiao et al., 2021*; *Zhu, Hou & Tong, 2021*; *Baudin et al., 2024*), our findings highlight critical gaps in radiation workers' knowledge and practices, underscoring the urgency of evidence-based interventions.

The KAB theory explains changes in human health-related behaviors by dividing them into three continuous processes as follows: acquiring knowledge, generating attitudes, and forming behavior. According to this theory, knowledge is the foundation, while attitude acts as the driving force behind behavioral change. DAGs provide a structured approach to visualizing these relationships, more intuitively identifying potential confounding variables, and obtaining effect estimates that are closest to the real-world outcomes (*Liu et al., 2019*). By integrating the KAB framework with causal inference through DAGs, this study advances beyond traditional association-based analyses to elucidate mechanistic pathways and confounding effects, providing actionable insights for policy and practice.

The significant direct effect of radiation protection knowledge on behavior ($\beta = 0.551$, $p < 0.001$) underscores its central role within the KAB framework. Our data reveal an "invisible risk" paradox where workers demonstrated strong behavioral compliance (mean = 28.99/35) despite critical knowledge gaps in biological effects (32% correct), radiation culture (28%), emergency protocols (25%), and dose limits (38%). This paradox operates through dual mechanisms supported by prior literature. For instance, heuristic dread of radiation's catastrophic potential triggers instinctive caution independent of technical knowledge (*Nyathi, 2022*; *Yurt et al., 2022*), while institutionalized safety rituals foster procedural compliance that typically decouples from substantive risk understanding (*Partap et al., 2019*; *Nyathi, 2022*). Consequently, 89% reported consistent equipment use, while 63% distrusted dosimeter accuracy—a disconnect reflecting "compliance without comprehension" that creates fragile adherence vulnerable to erosion during periods of high workload. Evidence-based interventions such as risk visualization tools, immersive simulations, and cognitive reframing are recommended to bridge this cognitive-behavioral gap (*Fujibuchi, 2021*). Our mediation analysis further reinforces this finding: the direct effect of knowledge on behavior far exceeded the indirect contribution of attitude ($\beta = 0.024$, 95% CI [0.003–0.045]), confirming that knowledge acquisition alone can drive behavioral improvements without attitudinal shifts. This finding partly challenges conventional KAB models, which position attitude as the primary mediator, implying that training programs should prioritize knowledge dissemination—particularly on underemphasized topics such as radiation culture, biological effects, and emergency protocols—to achieve rapid compliance.

For example, participants demonstrated a limited understanding of dose limits (mean knowledge score = 8.65/14) in radiation protection education, underscoring the need for curriculum standardization. Notably, the weak correlation between knowledge and attitude ($r = 0.256$, $p < 0.01$) shows that knowledge alone is insufficient to foster attitudinal change.

This disconnect reflects institutional or cultural barriers, such as skepticism toward the accuracy of the personal dosimeter (the lowest-scored attitude item). The lowest attitude score ($3.1 \pm 1.2$) was observed for "trust in personal dosimeter accuracy", with 63% of participants expressing skepticism. This finding aligns with concerns regarding third-party monitoring transparency (*Wrzesień, 2020*), as the hospital in our study relies on quarterly thermoluminescent dosimeter reports from external vendors, resulting in a 90-day feedback lag.

While 89% of participants reported the consistent use of protective equipment and patient guidance, 18% admitted non-compliance with dosimeter protocols. Multivariate analysis identified age ($\beta = -0.390$, $p < 0.001$) and radiation protection training ($\beta = 0.167$, $p < 0.001$) as key modifiers, with older workers demonstrating reduced compliance, regardless of knowledge levels.

Age, educational level, radiation protection education, number of children, knowledge of radiation protection, and attitude toward radiation protection were the primary factors influencing radiation protection behaviors among radiation workers.

Specifically, age independently predicted reduced protective behavior ($\beta = -0.390$, $p < 0.001$), with workers aged >50 years scoring significantly lower than those aged <30 years. This decline persisted even after adjusting for knowledge and attitude, implicating factors such as occupational burnout or risk habituation (*Chen & Gholamrezanezhad, 2023*).

Longitudinal studies indicate that prolonged engagement in high-volume clinical environments reduces vigilance among frontline staff due to occupational burnout or risk habituation (*Dyrbye et al., 2021*), corroborating our finding of an age-related behavioral decline ($\beta = -0.390$, $p < 0.001$). Older workers with longer career durations ($\geq 10$ years) exhibited significantly lower protective behavior scores ($p < 0.01$), likely due to cumulative burnout from repetitive high-risk tasks, which attenuates adherence to safety protocols (*Canon et al., 2022*).

We recommend age-specific interventions, including ergonomic adjustments, mental health screenings, and shorter shifts for senior personnel, to mitigate this effect. These measures align with broader occupational health strategies advocating for tailored support in aging workforces (*Parikh, 2023*).

Although educational level significantly correlated with behavior in the univariate analysis ($p = 0.001$), DAG analysis identified it as a confounder ($\beta = 0.101$, $p = 0.029$). Higher education enhanced knowledge acquisition but did not directly improve compliance. This finding explains why participants with a bachelor's degree (or higher education level) scored 15% higher in radiation protection behaviors, likely due to greater access to more sophisticated knowledge resources. However, these workers still required contextual training to translate knowledge into action.

Therefore, policy interventions should be stratified by educational level. Workers with lower educational attainment benefit from simplified, hands-on training, while degree holders require advanced modules on emerging risks—including low-dose chronic effects—to bridge the knowledge-behavior gap (*Yashima & Chida, 2022*; *Alkhayal et al., 2023*).

Radiation protection training showed a dual efficacy, enhancing both knowledge ($\beta = 0.394$, $p < 0.001$) and behavior ($\beta = 0.147$, $p < 0.001$). However, its limited impact on attitude indicates that current programs overlook attitudinal components, such as fostering a radiation culture or peer accountability. Therefore, integrating scenario-based simulations (*e.g.*, mock emergencies) or mentorship systems may address this gap (*Partap et al., 2019*; *Yashima & Chida, 2022*; *Salih et al., 2023*; *Shafiq & Mehmood, 2024*). While the model explained 51.6% of protective behavior variance ($F = 30.655$, $p < 0.001$), unmeasured confounders such as family planning significantly influenced outcomes. Workers without children scored 12% higher in protective behaviors than those with two or more children ($p = 0.032$), possibly due to divergent risk perceptions. Specifically, workers without children may prioritize self-protection due to fertility concerns, as ionizing radiation exposure is linked to impaired spermatogenesis (*Kesari, Agarwal & Henkel, 2018*; *Wdowiak et al., 2019*), whereas those with multiple children may perceive reduced reproductive risks, leading to complacency. This inverse relationship ($\beta = 0.136$, $p = 0.032$) underscores the need for tailored messaging emphasizing familial health risks beyond individual concerns. Training emerged as a critical modifiable factor, with participants who received regular training scoring 17% higher than those who did not attend training ($p < 0.001$). This aligns with previous research demonstrating the dual role of training in enhancing knowledge and compliance (*Yashima & Chida, 2022*; *Baudin et al., 2024*).

However, the knowledge-behavior gap persisted. Workers with two or more children scored 8.65/14 on knowledge assessments yet exhibited lower compliance, indicating motivational barriers. Future studies should adopt mixed-method approaches to explore unmeasured determinants such as workplace culture and risk communication efficacy. Furthermore, expanding the sample to include multilevel hospitals and a broader range of healthcare professionals would enhance generalizability.

Longitudinal study designs could further clarify temporal relationships between training exposure, attitude formation, and sustained behavioral change. Future studies should prioritize implementing multi-site studies across diverse hospital settings (*e.g.*, teaching *vs.* non-teaching, urban *vs.* rural), integrating electronic health records with real-time dosimetry to objectively track compliance behaviors while overcoming self-report biases. These trials should embed mixed-methods components—combining longitudinal dose measurements with periodic focus groups—to examine contextual mediators like organizational culture. Addressing these gaps will allow interventions to move beyond simple knowledge transfer to foster risk-perception alignment and lasting safety practices.

## Study limitations

This study has some limitations. First, its cross-sectional design precludes causal inference; however, the use of DAGs strengthens the theoretical plausibility. Second, the use of convenience sampling at a single non-teaching hospital may introduce selection bias (*e.g.*, overrepresentation of compliant workers attending routine safety meetings), limiting the findings' generalizability to teaching hospitals or other large comprehensive hospitals with different workflows. Third, self-reported data carry the risks of recall and social desirability biases, particularly for sensitive topics such as non-compliance. Although we ensured

anonymity and emphasized honest responses to mitigate these biases, participants may still have overreported protective behaviors due to perceived social expectations. Future studies should incorporate objective measures, such as electronic dosimeter usage logs or direct observation, to validate self-reported practices. Finally, departmental sample size disparities (*e.g.*, radiology $n = 113$ *vs.* endoscopy $n = 28$) may skew subgroup comparisons.

## CONCLUSIONS

This cross-sectional study of 335 healthcare workers across western China systematically evaluated radiation protection behaviors using a DAG to clarify causal pathways. Key findings demonstrate that radiation protection knowledge directly drives protective behavior, with age and radiation protection training as critical modifiers. DAG analysis identified educational level and the number of children as confounders, highlighting the need for tailored interventions.

Notably, workers aged ≤30 years, with bachelor's degrees, and without children exhibited higher compliance rates than the others. This study validates the utility of DAGs in disentangling complex relationships, revealing that knowledge acquisition alone explains 51.6% of behavioral variance, thereby challenging traditional KAB model assumptions.

Multivariate analysis confirmed significant associations between protective behavior and age, radiation protection training, and fertility status; however, the DAG highlighted unmeasured confounders such as workplace culture. We suggest that future research could adopt a mixed-method design to measure or account for such complex factors. For instance, combining longitudinal surveys with ethnographic observations and conducting structured interviews to explore team responsibility norms, thereby enabling clear adjustments to these latent factors in the causal model. These findings underscore the need for standardized training curricula emphasizing emergency protocols, dose limits, and age-specific ergonomic adjustments.

Furthermore, this research provides causal evidence supporting the prioritization of knowledge dissemination and system transparency in radiation protection. By integrating DAG analysis and KAB theory, our findings advance occupational health methodology and inform policy strategies to improve healthcare workers' safety.

## ACKNOWLEDGEMENTS

We would like to thank all participants in this study. This revised manuscript has been professionally edited by Editage to ensure grammatical accuracy and clarity.

### Funding

This research was funded by the Hospital level project of Mianyang Central Hospital, grant number 2023YJ001. The funders had no role in study design, data collection and analysis, decision to publish, or preparation of the manuscript.

## Grant Disclosures

The following grant information was disclosed by the authors:
Hospital level project of Mianyang Central Hospital: 2023YJ001.

## Competing Interests

The authors declare there are no competing interests.

## Author Contributions

- Lulu Cao conceived and designed the experiments, performed the experiments, analyzed the data, prepared figures and/or tables, and approved the final draft.
- Jinwen Wu conceived and designed the experiments, performed the experiments, analyzed the data, prepared figures and/or tables, and approved the final draft.
- Li Jiang performed the experiments, prepared figures and/or tables, and approved the final draft.
- Yi Tao performed the experiments, prepared figures and/or tables, and approved the final draft.
- Huaping Huang conceived and designed the experiments, prepared figures and/or tables, authored or reviewed drafts of the article, and approved the final draft.
- XiaoJing Xue conceived and designed the experiments, prepared figures and/or tables, authored or reviewed drafts of the article, and approved the final draft.

## Human Ethics

The following information was supplied relating to ethical approvals (i.e., approving body and any reference numbers):

The Biomedical Ethics Committee of Mianyang Centre Hospital granted Ethical approval to carry out the study within its facilities (S20240372-01).

## Data Availability

Data is available in the Supplemental File.

## Supplemental Information

Supplemental information for this article can be found online at http://dx.doi.org/10.7717/peerj.20083#supplemental-information.

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
