# Peer review of "Exploring influences on radiation protection compliance: a directed acyclic graph-based cross-sectional study in a non-teaching hospital in western China"

_PeerJ, doi:10.7717/peerj.20083_

## Round 0.1 · original submission · Major Revisions

The authors have investigated a crucial topic in occupational health through the use of an intriguing methodological approach (DAGs). The creative application of DAGs for causal inference and the discovery of useful knowledge gaps are among the advantages. The manuscript does, however, need extensive linguistic improvement and clarification throughout. According to journal policy, the availability of raw data must be addressed (public accessibility vs. contact-on-request). Additionally, the debate should go deeply into the implications of the partial challenge of the KAB paradigm and offer more specific suggestions for overcoming particular attitudinal barriers. The paper has the potential to be a significant contribution to the field with these significant changes.

**PeerJ Staff Note**: Please ensure that all review, editorial, and staff comments are addressed in a response letter and that any edits or clarifications mentioned in the letter are also inserted into the revised manuscript where appropriate.

**Language Note**: The review process has identified that the English language must be improved. PeerJ can provide language editing services - please contact us at [email protected] for pricing (be sure to provide your manuscript number and title). Alternatively, you should make your own arrangements to improve the language quality and provide details in your response letter. – PeerJ Staff

·

Basic reporting

Substantially improve the English language throughout the manuscript. Numerous grammatical and stylistic errors hinder clarity and must be corrected.

Revise Figure 1 to enhance readability. The current version is too small, and the labels are difficult to read. Provide a higher-resolution image with clearer node and edge labels. Make the dashed lines representing "potential indirect effects" more visually distinct.

Expand the discussion on the KAB framework. Provide deeper analysis and contextual relevance to strengthen the interpretation of findings.

Experimental design

-

Validity of the findings

Methods Section:

Questionnaire Scales: While Cronbach's alphas are provided, more detail on the development and validation process of the "validated scales" would be beneficial for replication. Simply stating they were based on existing research and reviewed by experts is a start, but further detail on how content validity, face validity, and construct validity were ensured.

DAG Construction Details: Line 203 mentions "DAGitty package (version 3.0) in R Studio." While this is good, briefly explain how DAGitty was used to verify logical consistency (e.g., by checking for d-separation or identifying adjustment sets). This will make the DAG analysis more transparent for readers less familiar with the tool.

Sampling Method Justification: "Convenience sampling" (Line 137) is noted. Briefly discuss the potential biases introduced by this non-probability sampling method in the "Limitations" section, in addition to the single-center aspect.

Results Presentation:

Table 1: The "t/F P" column contains P values for both t-tests and F-tests (ANOVA). It would be helpful to explicitly state which test was used for each row if it's not immediately obvious from the "Categories" (e.g., gender is 2 categories, profession has multiple). Also, the interpretation of the P-values for categorical variables (e.g., Age, Education Level) refers to differences in the mean protective behavior score (M ± SD), which is implicitly the dependent variable in Table 1, but this should be explicitly stated in the table description or a footnote to avoid confusion.

Table 4: The R2 and Adjusted R2 values are provided below the table, which is good. Ensure consistent decimal places for all values (B, SE, Beta, t, P). For instance, some P-values are given as "ÿ0.001", which should be "<0.001".

Table 5: This table is crucial. Ensure consistent decimal places and clear notation for p-values (e.g., "<0.001"). The meaning of "Standardized β" for Indirect Effect (Knowledge → Attitude → Behavior) needs a brief explanation in the table caption or text, especially since it's a mediated effect.

Discussion Points:

"Invisible Risk" Paradox: Expand on the "invisible risk" paradox (Lines 290-291). How might this paradox influence compliance even with positive attitudes, and what interventions are effective in addressing such risks?

Unmeasured Confounders: The discussion mentions "unmeasured confounders (e.g., workplace culture)" (Line 264, 346, 388). While acknowledged as a limitation, briefly suggest how future research could attempt to measure or account for such complex factors (e.g., qualitative studies, organizational surveys).

Future Research: The conclusion states, "Longitudinal designs could further clarify temporal relationships" (Lines 364-365). This is a strong suggestion and should be emphasized. Additionally, suggest specific methodological avenues for future studies to address the limitations identified (e.g., mixed-methods, multi-site studies).

Reviewer 2 ·

Basic reporting

Figures are relevant to the content of the work. However, Figure 1 may require a larger font size to be readable in the final publication (See Note 13).

Experimental design

The findings are neither unique nor unexpected. However, the authors are reporting results from a medium-sized hospital in western China. It is of interest to the radiation protection community to understand how effective compliance with radiation protection standards is across the globe.

Several unfamiliar statistical terms are used without explanation (see Notes 6, 7, and 10).

The work was performed in a rigorous fashion. However, there is a question regarding the wording of one section of the questionnaire administered to the study participants. As presented in the file, seven questions seem to ask the participant to promise to behave properly in the future (begins with “I will”) (See Note 4). Unless there is an error in translating the questionnaire into English, the structure of these seven questions could invalidate several stated results and several key conclusions.

Validity of the findings

Conclusions were well constructed. However, due to the concerns expressed above and in Note 4, portions of the Discussion and Conclusion section may need to be rewritten.

Additional comments

No. Section Line No(s). Comments/Recommendations
1. Introduction, Lines 73-86: These two paragraphs contain unnecessary background information. Recommend: delete.

2. Materials and Methods, Section 2.1, Lines 136: It is not clear what “Level III general hospital” means. Does this hospital serve a local, regional, or provincial area?

3. Materials and Methods, Section 2.1, Lines 139: The term “radiologist” is generally restricted to physicians who are board-certified in Radiology. Is this term used more widely in China?

4. Medical staff radiation protection knowledge and practice questionnaire: Questions 20-26 begin with the phrase “I will.” This implies that individuals completing the questionnaire are making a promise about behavior in the future. It does not appear to assess individuals’ current behavior. This calls into question the validity of any conclusions drawn about behavior. If this interpretation is correct, the authors must rewrite the manuscript with this in mind.

5. Materials and Methods, Section 2.2, Line 150: What was the expertise of the “six experts”? What criteria were used in their selection?

6. Materials and Methods, Section 2.2, Line 166: Add reference for “Likert scale.”

7. Materials and Methods, Section 2.2, Line 171: Add reference for “Cronbach’s alpha coefficient”

8 Materials and Methods: Behavior was assessed by having the participants answer questions on a form. This is a highly subjective process and could be subject to responders’ biases and concerns for giving an answer that is “correct” versus how they actually behave in practice. Was this/how was this issue addressed in the data analysis? If it was not addressed in the data collection and analysis process, the authors should consider discussing this potential issue in the Discussion.

9 Results, Section 3.5, Line 253. The statement about “direct effects of knowledge” is unclear. Is it reasonable to assume that the “direct effects of knowledge” means that those participants with more demonstrated knowledge performed better in compliance? Please clarify.

10. Results, Section 3.5, Line 263: Identify/define “DAGitty package”

11. Discussion, Lines 283-298. This paragraph begins with a statement about knowledge emerging as the strongest predictor of protective behavior and gives statistics. The sentence ends with a citation of two publications from the literature. This appears to imply that the data stated in the sentence comes from the two cited publications. However, the data presented in this sentence and others in this paragraph come from the present work. Please clarify that the data presented and discussed in this paragraph come from the present work.

12. Discussion, Lines 320-322: The statement “Longitudinal studies indicate that prolonged radiation exposure reduces vigilance among 321 frontline staff managing high patient volumes…” is unclear. This reviewer presumes that the authors do not mean that prolonged exposure to radiation reduces vigilance, but that a long history of work in a busy environment may reduce vigilance, perhaps due to complacency or apathy. Please reconcile.

13. Figure 1: Font size may need to be increased for final publication.

---

## Round 0.2 · accepted · Accept

Recommend acceptance for publication in PeerJ.

Reviewer 2 ·

Basic reporting

No additional comments beyond those previously submitted.

Experimental design

No additional comments beyond those previously submitted.

Validity of the findings

No additional comments beyond those previously submitted.

Additional comments

All concerns stated by this reviewer have been address appropriately. Recommend approval for publication.